# Bioactivity of Macronutrients from *Chlorella* in Physical Exercise

**DOI:** 10.3390/nu15092168

**Published:** 2023-04-30

**Authors:** Karenia Lorenzo, Garoa Santocildes, Joan Ramon Torrella, José Magalhães, Teresa Pagès, Ginés Viscor, Josep Lluís Torres, Sara Ramos-Romero

**Affiliations:** 1Physiology Section, Department of Cell Biology, Physiology and Immunology, Faculty of Biology, University of Barcelona, 08028 Barcelona, Spain; 2Department of Biological Chemistry, Institute of Advanced Chemistry of Catalonia (IQAC-CSIC), 08034 Barcelona, Spain; 3Laboratory of Metabolism and Exercise (LaMetEx), Research Centre in Physical Activity, Health and Leisure (CIAFEL), Faculty of Sport, University of Porto, 4200-450 Porto, Portugal

**Keywords:** algae, fatty acid, protein, fiber, physical activity

## Abstract

*Chlorella* is a marine microalga rich in proteins and containing all the essential amino acids. *Chlorella* also contains fiber and other polysaccharides, as well as polyunsaturated fatty acids such as linoleic acid and alpha-linolenic acid. The proportion of the different macronutrients in *Chlorella* can be modulated by altering the conditions in which it is cultured. The bioactivities of these macronutrients make *Chlorella* a good candidate food to include in regular diets or as the basis of dietary supplements in exercise-related nutrition both for recreational exercisers and professional athletes. This paper reviews current knowledge of the effects of the macronutrients in *Chlorella* on physical exercise, specifically their impact on performance and recovery. In general, consuming *Chlorella* improves both anaerobic and aerobic exercise performance as well as physical stamina and reduces fatigue. These effects seem to be related to the antioxidant, anti-inflammatory, and metabolic activity of all its macronutrients, while each component of *Chlorella* contributes its bioactivity via a specific action. *Chlorella* is an excellent dietary source of high-quality protein in the context of physical exercise, as dietary proteins increase satiety, activation of the anabolic mTOR (mammalian Target of Rapamycin) pathway in skeletal muscle, and the thermic effects of meals. *Chlorella* proteins also increase intramuscular free amino acid levels and enhance the ability of the muscles to utilize them during exercise. Fiber from *Chlorella* increases the diversity of the gut microbiota, which helps control body weight and maintain intestinal barrier integrity, and the production of short-chain fatty acids (SCFAs), which improve physical performance. Polyunsaturated fatty acids (PUFAs) from *Chlorella* contribute to endothelial protection and modulate the fluidity and rigidity of cell membranes, which may improve performance. Ultimately, in contrast to several other nutritional sources, the use of *Chlorella* to provide high-quality protein, dietary fiber, and bioactive fatty acids may also significantly contribute to a sustainable world through the fixation of carbon dioxide and a reduction of the amount of land used to produce animal feed.

## 1. Introduction

Algae, as well as the products made from them, are in increasing demand worldwide because of their nutritional value and practical contributions. Microalgae (unicellular algae) are a source of high-quality proteins, similar to those found in milk, eggs, and meat, with a low fat content [1]. They also include other bioactive components, specifically polyunsaturated fatty acids (PUFAs), polysaccharide fibers, polyphenols, carotenoids, phycobiliproteins, vitamins, and sterols [2,3,4]. In addition to the nutritional benefits provided by these, the viability of cultivating microalgae in specific installations as an alternative source for CO_2_ fixation, thereby capturing this greenhouse gas without occupying soil, makes them particularly promising for the sustainability of the planet. Microalgae are used in the pharmaceutical, cosmetic, and food industries; however, the European Commission only lists two species in their Novel Food Catalogue: *Arthrospira* (also called Spirulina, a cyanobacterium) and *Chlorella* (a green alga). (In this paper, we use the terms “*Chlorella*” when referring to the genus of this living organism, and “spirulina” and “chlorella” when referring generically to biomass preparations of these microalgae. We will also use the terms “spirulina” and “chlorella” when referring to studies or reports in which the particular species used is not specified.)

*Chlorella* was first described by Beijerinck in 1890. Its name is derived from ‘chloros’ (from Greek, meaning green) and the suffix ‘ella’ (from Latin, meaning small). Nowadays, more than 20 species and over 100 strains of *Chlorella* have been described [5], belonging to 2 classes of Chlorophyta: Chlorophyceae and Trebouxiophyceae. *Chlorella* is a spherical/ellipsoidal cell with a 2–10 µm diameter that reproduces via asexual autospores. The main habitats where *Chlorella* lives are both fresh water and seawater, although it can also be found in soil, living independently as well as in symbiosis with lichens or protozoa [6]. The biochemical compositions of *Chlorella* vary greatly between species and even strains and also depend on the culture conditions, including nutritional and environmental factors. The general growth and protein production of this microalga increase with the rising nitrogen content of its medium, while nitrogen limitation augments the proportion of starch or lipids in *Chlorella* [6]. In general, the macronutrient composition of *Chlorella* biomass is over 60% dry weight of protein (including all the essential amino acids) and more than 10% of both lipids and carbohydrates (Table 1) [7]. Moreover, *Chlorella* contains many different components with functional activities (Table 1).

Spirulina is the other microalga approved for human consumption [4]. Preparations of chlorella and spirulina have different proportions of common components. The polysaccharide (β-glucans and arabinomannans) composition of *Chlorella* spp. has been better defined than that of Spirulina [4]. *Chlorella* contains β-glucans (polymers of β-D-glucose linked through 1–3 β-glycosidic bonds, 6–9% of dry weight) [8] and arabinomannans (oligomers of arabinose and mannose) [9] as well as other less known saccharides with hypolipidemic activity [10]. As well as these lipidemic functions, up to 50% of *Chlorella* dry weight can consist of triacylglycerols (TAGs) when it has been exposed to high light or nitrogen deficiency conditions [11]. Peptides from the enzymatic hydrolysis of *Chlorella* can also have health benefits, such as hypoglycemic, anti-inflammatory, and blood pressure-lowering activities (e.g., inhibition of angiotensin I-converting enzyme, ACE) [4,12,13]. Therefore, because of its nutritional value and bioactive compounds, *Chlorella* is an interesting microalga for human consumption; however, if it is not properly processed, it could be poorly digested and induce side effects because, in its naturally occurring form it contains a cellulose cell wall. So, chlorella must be mechanically broken down during production for human consumption to avoid gastrointestinal issues, including nausea, vomiting, and stomach problems. Other side effects related to chlorella intake include renal and allergy problems [14]. Despite this, in the context of human health and disease, chlorella has been shown to have cardiovascular benefits, as revealed by its capacity to lower total cholesterol, low-density lipoprotein cholesterol, systolic blood pressure, diastolic blood pressure, and fasting blood glucose in healthy individuals and patients with non-alcoholic fatty liver disease, hypertension, hypercholesterolemia, and dyslipidemia [15]. At the doses used to elicit these responses, chlorella is considered to be safe [16].

In addition to its macronutrients, chlorella also contains phenolic compounds with radical scavenging capacity and α-amylase/glycosidase inhibitory activity [12], as well as other components such as vitamins (e.g., vitamin B_9_ or folate) [17] and carotenoids (e.g., astaxanthin) [18] that may contribute to its antioxidant and anti-inflammatory activities [19]). Moreover, it incorporates chlorophyll and other minerals with some biological functions [7]. Taken together, it seems that the functional activity of chlorella intake could be complementary or synergistic with other health-related habits, such as consuming a balanced diet, getting adequate sleep, or taking physical exercise.

Physical exercise represents a stressful activity for the body’s cells, tissues, and organs, which dysregulates whole-body homeostasis in a progressive and reversible way. However, as a consequence of its regular and systematic practice, different cellular signaling pathways become activated and generate systemic and local chronic adaptations. These promote physiological, biochemical, and morphological adjustments of the organism to the exercise [20], such as a gain of muscle mass and strength and improvements in cardiovascular health and functional capacity [21].

Besides their role as biochemical triggers and mechanical platforms for strength production, skeletal muscle fibers are nowadays also considered to be essential pleotropic sources of several compounds and molecules crucial for muscle-to-organ/tissue cross-talk communication [22]. Myokines, for example, are specific cytokines, small peptides, growth factors, and metallopeptidases produced and released by skeletal muscle, with both paracrine and endocrine effects, that promote communication between muscle cells and other target organs or tissue cells [21]. It has been shown that regular exercise increases the production and secretion of myokines, such as irisin, insulin-like growth factor 1 (IGF-1), brain-derived neurotrophic factor (BDNF), meteorin-like protein (Metrnl), fibroblast growth factor, β-aminoisobutyric acid (BAIBA), the interleukins (ILs) IL-15, IL-7, IL-6, and decorin, whilst reducing the production and release of myostatin. These myokines are the drivers of the most beneficial effects of exercise in terms of health-related outcomes and chronic adaptive responses to training [23,24]. This is because their regulation is directly related, among other things, to increased protein synthesis and reduced muscle protein breakdown, increased lipid oxidation, browning of white adipose tissue, increased insulin sensitivity, increased myogenesis, and satellite cell activation (for a review, see [25]). Furthermore, it is known that, both during and after conventional exercise models, contracting muscles increase ROS production in a way that positively impacts muscle cellular signaling and adaptive responses, thereby strengthening cells to deal better with the demands of physical exercise and to mitigate several deleterious consequences associated with pathological conditions [20,22]. Regarding health-related benefits, regular exercise has been defined as a prophylactic but also a therapeutic, non-pharmacological “polypill”, able to prevent around 26 different chronic diseases and reduce all-cause mortality [23,24,26], by reducing the incidence of cardiovascular diseases; through the regulation of blood lipids, hypertension, and arterial fitness [22]; and by protecting against atherosclerosis, type 2 diabetes, and breast and colon cancer [23]. Moreover, regular exercise is a consensual treatment tool against different pathological risk factors or situations, such as obesity (by helping practitioners to lose weight, reducing the metabolic risk factor, and improving adipose tissue health [22,27]), ischemic heart disease, heart failure, type 2 diabetes, chronic obstructive pulmonary disease, chronic low-grade systemic inflammation, and non-alcoholic fatty liver disease [23]. Furthermore, physical exercise has been considered crucial to mental health, helping in the management of depression and anxiety and in the therapeutic challenge against neurodegenerative disorders such as Alzheimer’s or Parkinson’s disease [28,29]. Apart from preventing and treating pathological situations, regular physical exercise can improve cardiovascular fitness and aerobic capacity, thereby slowing biological aging through a reduction of age-related sarcopenia and thus increasing quality of life and potentially life expectancy.

Chlorella, as well as spirulina and other microalgae, is progressively being introduced into human and animal nutrition as a wholesome source of nutrients that is complementary or an alternative to animal-derived foodstuffs. Whereas research on the properties of edible microalgae in the context of health and disease has been summarized many times, the important area of sports nutrition has not been systematically reviewed to date. Therefore, the present paper aims to contribute to the advancement of research on the use of new environmentally friendly and health-promoting foodstuffs in this area. Taken together, it is reasonable to assume that microalgae and their bioactive compounds are interesting nutritional sources to improve exercise performance and post-exercise recovery, as well as enhance the beneficial clinical features of physical exercise [1]. This paper reviews the available evidence supporting the benefits of the edible microalgae *Chlorella* in the context of physical exercise; however, the role played by chlorella’s multiple components in these effects is still an active area of research.

## 2. Methods

We searched the literature using the PUBMED and Google Scholar databases in June 2022. A variety of combinations of words and terms were entered as the search criteria, including, but not limited to, “chlorella”, “physical activity”, “training”, “exercise”, “algae”, “fatty acid”, “protein”, “fiber”, “polysaccharides”, and “PUFAs”.

Both animal and human studies are included in this review. Additionally, healthy subjects as well as animals or humans with any stated medical condition are included. Manuscripts published before 1992 and non-English language publications were excluded. We also excluded from the analysis studies with no exercise protocol or those with no direct or indirect relationship with physical exercise.

Nineteen papers that evaluated supplementation with chlorella or some of its macronutrients in the context of physical exercise are included in this review. Some features of these articles that discuss chlorella supplementation, such as participants, supplementation, exercise protocols, and findings, are summarized in Table 2.

## 3. *Chlorella* spp. and Physical Exercise

The use of chlorella as a nutritional source of specific bioactive compounds in the context of novel foods or as a supplement has been shown to enhance physical exercise performance in humans and other mammals (Table 2). For example, through the improved expression of monocarboxylate transporters (MCT) 1 and 4, peroxisome proliferator-activated receptor gamma (PPAR-γ) coactivator-1α, and the activities of lactic dehydrogenase (LDH), phosphofructokinase, citrate synthase, and cytochrome oxidase (COX) in the red region of rat muscle, chlorella synergistically enhances the ability to perform maximum numbers of high-intensity intermittent training (HIIT) sessions as a result of training [34]. These biochemical effects increased the ability to produce ATP in skeletal muscle via the glycolytic and oxidative pathways, resulting in a significant improvement in exercise performance. Additionally, diverse chlorella preparations can influence the magnitude of intense exercise-related muscle damage and physical stamina. Hot water extracts from *Chlorella vulgaris* decrease blood urea nitrogen (BUN), creatine kinase, and LDH in the blood serum of mice after a forced swimming test [30]. Additionally, hydrolyzed *C. vulgaris* reduces the immobility time in forced swimming tests in mice while reducing BUN [31]. Despite numerous studies that link chlorella intake with improvements in different aspects of physical exercise, there is still a lack of consensus about the main mechanism(s) via which these effects are produced. In general, chlorella intake has been shown to have mainly antioxidant, immunomodulatory, metabolic, and antihypertensive effects [7]. All together, these effects could act synergistically to improve exercise performance.

The capacity of the bioactive compounds present in *Chlorella* to scavenge free radicals could be one of the main mechanisms underlying its biological activities. *C. vulgaris* supplementation improves several oxidative stress markers, such as malondialdehyde (MDA), total antioxidant capacity (TAC), and lipid peroxidation in rats in which oxidative stress has been induced [37], as well as increasing the activities of some antioxidant enzymes, such as superoxide dismutase (SOD) and catalase (CAT) in smokers [38]. Additionally, some of its bioactive compounds, like astaxanthin, modulate signaling pathways related to oxidative stress, such as those involving mitogen-activated protein kinases (MAPK), phosphatidylinositol-3 kinase/protein kinase B (PI3K/Akt), and nuclear factor erythroid 2-related factor 2/antioxidant response element (Nrf2/ARE) [39]. Altogether, the data suggest that *Chlorella* has potential as an antioxidant during and after physical exercise. However, the role of antioxidants in exercise training is still controversial. Antioxidant supplementation can reduce the advantages induced by training, as some chronic exercise adaptations related to performance are mediated by both ROS and reactive nitrogen species (RNS), namely enhancements in antioxidant capacity, mitochondrial biogenesis, cellular defense mechanisms, and insulin sensitivity [40]. In general, it seems that physiological doses of ROS/RNS are beneficial when it comes to improving exercise performance-related “machinery”, while exaggerated production may be detrimental. Therefore, more studies are needed to ascertain adequate chlorella doses that support antioxidant benefits related to exercise intensity and duration without compromising ROS/RNS-related cellular signaling adaptations and, ultimately, performance.

Similarly to the production of ROS/RNS, physical exercise also increases the concentration of a number of pro-inflammatory cytokines in the blood [23], depending on the intensity, duration, and predominant mode of the imposed muscle contractions. Specifically, IL-6 is the first cytokine present in the bloodstream during physical exercise; in turn, it activates the production of other cytokines, such as IL-1 and IL-10, and inhibits the release of TNF-α. Data regarding *Chlorella vulgaris* show that it reduces the production of IL-6 and TNF-α in vitro, as well as reducing NO and prostaglandin E2 (PGE_2_) release, which suggests that this microalga has significant anti-inflammatory activity [41]. The activity of *Chlorella vulgaris* on serum IL-6 levels has also been related to a reduction of insulin resistance after one session of intense acute eccentric exercise in overweight men [36]. This ability of chlorella to modulate insulin signaling pathways seems to also be related to its activity on muscle sirtuin 1 (SIRT1) [34], as increased expression of SIRT1 improves insulin sensitivity and attenuates insulin resistance [42]. A combination of chronic intake of chlorella and endurance training further increased PI3K activity, Akt phosphorylation, and glucose transporter type 4 (GLUT4) translocation in the skeletal muscle of diabetic rats [35]. This combined treatment may produce larger beneficial effects on improving glycemic control via the enhancement of skeletal muscle glucose uptake.

The effects of chlorella on glucose metabolism may also be interesting for exercise performance under several specific conditions. During moderate to intense or prolonged exercise, liver glycogen stores and gluconeogenesis help to maintain blood glucose levels in the body and sustain the glycolytic flux in the active musculature. Surprisingly, chlorella intake preserved liver glycogen stores and performance during a swimming test, as it prompted an energy shift from carbohydrate to free fatty acid utilization [32].

Glucose metabolism is linked to blood pressure as insulin resistance is a main upstream event leading to hypertension [43]. However, there are other factors that can also modulate blood pressure, such as nitric oxide (NO) levels. Local NO production by the vascular endothelium improves the flow-mediated dilation (FMD) that affects arterial function by decreasing peripheral vascular resistance, thus impacting general vascular health [44]. Chlorella supplementation increases NO production [45] and the expression of endothelial nitric oxide synthase (eNOS) in men, thereby augmenting blood flow to muscles [46], which could eventually enhance performance. The vasodilatation induced by NO would lead to greater muscle perfusion and O_2_ delivery, theoretically speeding up VO_2_ kinetics and improving muscle performance and recovery [47]. Accordingly, supplementation with chlorella leads to a significant increase in peak oxygen uptake, suggesting that chlorella increases endurance in young individuals [33]. Moreover, the antihypertensive effects of chlorella are significant not only in healthy young subjects but also in middle-aged and elder individuals, as reported by Otsuki et al. [45,48]. The antihypertensive effect of chlorella seems to be related to the activation of NO production by endothelial cells, which triggers vasodilatation, as well as the improvement of glucose metabolism.

Although the beneficial effects of chlorella on exercise performance might involve synergies between its bioactive features, they might also be related to some specific bioactive compounds that are present in it. So, discriminative knowledge of the benefits of each *Chlorella*-derived product as a positive ergogenic effector could be an interesting subject for the functional food industry, as well as for the consumer.

## 4. Bioactive Macronutrients from *Chlorella* and Physical Exercise

### 4.1. Benefits of Proteins and Peptides from Chlorella for Physical Exercise

*Chlorella* species have a great potential to be used as an alternative protein source because over 80% of them are digestible by humans and comparable both quantitatively and qualitatively to other conventional vegetable proteins [49]. In terms of quantity, proteins range from 51% to 58% in *C. vulgaris*, and 57% for *C. pyrenoidosa* on a dry weight basis in growth conditions that are rich in nitrogen. In the absence of nitrogen, the amount of proteins present can decrease by up to 20% *w*/*w* on a dry weight basis [49,50,51]. The current trend of increasing atmospheric CO_2_ may also lead to an increase in the protein content of these species [51]. In terms of quality, most microalgae contain all of the essential amino acids that mammals are unable to synthesize and therefore must obtain from their food intake [50]. Specifically, the essential amino acid index (EAAI) of *C. pyrenoidosa* indicates that all essential amino acids are present at substantial concentrations [52]. Compared to other macronutrients, protein intake increases satiety, activation of the anabolic mTOR (mammalian target of rapamycin) pathway in skeletal muscle, and the thermic effects of meals [53]. As plant-based diets seem to be a good option for enhancing athletic performance while also improving general physical and environmental health [54], a regular intake of *Chlorella* or its isolated proteins could be an environmentally friendly choice of dietary supplement for people undergoing training.

Apart from its protein content, *Chlorella* also contains bioactive peptides that may be of interest in terms of exercise performance. Bioactive peptides usually contain 3–16 amino acid residues, and their activities are mainly based on their amino acid composition and sequence [55]. Some bioactive peptides are antioxidants, as they scavenge ROS and free radicals or prevent oxidative damage by interrupting the radical chain reaction of oxidation. This antioxidant activity of peptides mainly depends on the hydrophobic amino acids they contain, specifically some aromatic amino acids and their histidine content [56]. A peptide from *C. ellipsoidea* (LNGDVW, 702.2 Da) has demonstrated great efficiency in scavenging various free radicals in vitro and therefore has the potential to be a good dietary supplement for the prevention of oxidative stress [57]. Another peptide from *C. vulgaris* (VECYGPNRPQF, 1309 Da) can efficiently quench a variety of free radicals, including hydroxyl, superoxide, peroxyl, 2,2-diphenyl-1-picrylhydrazyl (DPPH), and 2,2’-azino-bis(3-ethylbenzothiazoline-6-sulfonic acid) (ABTS) radicals, and also has significant protective effects on DNA and against cellular damage caused by hydroxyl radicals [58]. So, the intake of certain peptides from *Chlorella* with a tested antioxidant capacity could improve the detrimental oxidative stress potentially induced by particular high-intensity bouts of exercise or periods of exacerbated exercise regimens and training schedules [40].

Most analyses show that the highest proportions of amino acids in marine algae, and specifically in most *Chlorella* species, are glutamic and aspartic acids [6,59]. Besides the organoleptic properties of glutamate (one of the main components of the savory flavor, contributing to the basic umami taste), it also has bioactive activities that are potentially interesting in the context of exercise. Dietary supplementation with glutamic acid increases intramuscular free amino acid (FAA) concentrations and decreases the mRNA levels of genes involved in protein degradation in the skeletal muscle of growing pigs [60]. Moreover, when combined with leucine, it improves the FAA profile and mRNA levels of amino acid transporters in muscle, including neutral amino acid transporter 2 (ASCT2), large neutral amino acid transporter (LAT1), sodium-coupled neutral amino acid transporter 2 (SNAT2), low-affinity intestinal transporter of glycine and imino acids (PAT-1), and high-affinity renal transporter of glycine, proline, and hydroxyproline (PAT2) [61]. Meanwhile, aspartate, the other main amino acid in chlorella, enhances the ability of muscle to utilize free fatty acids during moderate exercise, thereby sparing glycogen and improving the biochemical capacity of the muscle for the oxidation of fatty acids through β-oxidation [62]. Chlorella products also contain high levels of arginine, a pivotal amino acid for the production of NO and the regulation of the immune system [7]. L-arginine supplementation in combination with other components improves tolerance of aerobic and anaerobic physical exercise in untrained and moderately trained subjects [63]. Moreover, supplementation with L-arginine together with physical training seems to be an important stimulus that induces significant improvements in exercise performance and redox status in rats [64].

Numerous papers have provided evidence that *Chlorella*-derived bioactive peptides have notably beneficial effects on human health. This evidence suggests that various amino acids and their metabolites play important roles in the skeletal muscle and have a significant overall ergogenic effect on exercise.

### 4.2. Benefits of Polysaccharides from Chlorella for Physical Exercise

A key part of the bioactivity of chlorella might be associated with its polysaccharides [65]. Polysaccharides with different structural features from various *Chlorella* species present a variety of biological activities, such as immunomodulatory, antioxidant, hypolipidemic, antitumor, or anti-asthmatic [66].

Research into the bioactivity of polysaccharides extracted from *C. pyrenoidosa* has focused on their effects on immunity and their antioxidant activities [66]. The polysaccharide fractions from *C. pyrenoidosa*, consisting mainly of the d-arabinose, d-glucose, d-xylose, d-galactose, d-mannose, and l-rhamnose moieties, have in vitro antitumor [67] and antioxidant capacity, specifically against superoxide and hydroxyl radicals [66]. α and β-Glucans, polysaccharides present in C. *pyrenoidosa* and *C. sorokiniana* [65], can modulate ROS production, the expression of the ROS-generating enzyme dual oxidase 2 (DUOX-2), and the immune factors TNF-α, IL-1β, and COX-2, thereby contributing to maintaining low levels of oxidative stress and pro-inflammatory molecules [68].

Dietary fiber consists of complex polysaccharides that are poorly or non-digestible by humans. High-fiber diets improve glycemic control, blood lipids, body weight, and inflammation [69]. *C. vulgaris* contains dietary fiber in highly variable proportions, ranging from 5.6% to 26.0% *w*/*w* on a dry weight basis [70]. In *Chlorella*-derived products, more than 65% of their carbohydrate content is dietary fiber from the *Chlorella* cell wall [7]. Soluble dietary fiber is mainly composed of resistant or functional oligosaccharides, such as fructo-oligosaccharides (FOS), galacto-oligosaccharides (GOS), and inulin, and viscous dietary fibers with a high molecular weight (glucan, pectins, and gums) [71]. Several studies suggest that both resistant oligosaccharides and viscous dietary fibers can effectively increase the bacterial diversity of the human gut microbiota and the abundance of bifidobacteria, lactobacilli, Prevotellaceae, and *Faecalibaculum* spp. [72]. Recent research has revealed a connection between the profile and diversity of the gut microbiota and the host’s physical performance [73]. Fiber intake promotes gut microbial diversity and increases the proportion of species, such as *Faecalibacterium prausnitzii*, *Lactobacillus* spp., *Bifidoacterium* spp., Firmicutes, and Bacteroidetes, that produce SCFAs [74]. SCFAs, including acetate, butyrate, and propionate, exert a wide range of metabolic functions, including anabolic regulation, insulin sensitivity, and modulation of inflammation, when absorbed into systemic circulation [75]. In vitro, *C. pyrenoidosa* increases the abundance of the genera *Prevotella*, *Ruminococcus,* and *Faecalibacterium*, as well as the production of butyrate and propionate [76]. Meanwhile, *C. pyrenoidosa* polysaccharides (CPPs) reduce *Escherichia-Shigella*, *Fusobacterium,* and *Klebsiella* and increase *Parabacteroides*, *Phascolarctobacterium,* and *Bacteroides* [77]. At the phylum level, CPPs increase the abundance of Bacteroidetes, leading to a lower Firmicutes/Bacteroidetes ratio [77], which seems to be closely related to body weight control and to the maintenance of intestinal barrier integrity, as precursors of low-grade inflammation, which are both important aspects affecting exercise performance. Other *Chlorella* species, such as *C. vulgaris* and *C. protothecoides,* increase propionate-producing bacteria in vitro [78]. In rats, a CPP treatment increased the growth of *Coprococcus*, *Turicibacter,* and *Lactobacillus* and the concentrations of acetate, propionate, and butyrate [10,79]. *C. vulgaris* also modulates *Lactobacillus* spp. metabolism, thereby increasing growth and the production of l-lactic acid while reducing the production of d-lactic acid [79]. Increasing the proportion of *Lactobacillus plantarum* in the gut seems to augment muscle mass, enhance energy harvesting, and have health promotion, performance improvement, and anti-fatigue effects [80].

Other aspects of physical performance can also be directly improved by SCFAs [73]. SCFAs can be used as carbon and energy sources for liver and muscle cells, thus improving endurance performance by maintaining blood glucose levels [81]. SCFAs also appear to regulate neutrophil function and migration, inhibit inflammatory cytokines, and control the redox environment in cells, which may help to enhance muscle renewal and adaptability, improve exercise performance, and delay symptoms of fatigue [81]. However, the efficacy of chlorella supplementation depends on the gut microbiota of the host [82], so future studies should focus on the underlying mechanisms implicated in the crosstalk between *Chlorella* polysaccharides, gut microbiota, and exercise performance in order to maximize the metabolic benefits of polysaccharides from *Chlorella*.

### 4.3. Benefits of Fatty Acids from Chlorella for Physical Exercise

*Chlorella* synthesizes high levels of unsaturated fatty acids in response to some environmental factors, such as temperature, pH, light, air composition (mainly nitrogen and phosphorous limitations), salinity, and nutrients [83]. The lipid composition of *C. vulgaris* includes about 23–34% of saturated fatty acids (SFAs), 15–25% of monounsaturated fatty acids (MUFAs), and 42–62% of PUFAs (of which over 30% are ω3 PUFAs) by weight as a percentage of the unfractionated total lipids [84,85]. The major PUFAs in *C. vulgaris* are linoleic acid (LA, C18:2 ω6; over 23%) and alpha-linolenic acid (ALA, C18:3 ω3; over 21%) [84]. LA is considered an essential fatty acid because higher animals, including humans, cannot synthesize it [86]. The proportions of total unsaturated fatty acids and PUFAs in *C. vulgaris* are the highest of the green microalgae [84], and they can be increased under favorable growing conditions. This makes it suitable for industrial and nutritional purposes [87].

Incorporating specific nutrients or dietary supplements to enhance exercise performance and recovery is a common strategy used by recreational practitioners and, particularly, by professional athletes. Moreover, supplementation with unsaturated fatty acids has been shown to have a number of biological effects on health and is related to diseases [88]. Many studies have evaluated the impact of unsaturated fatty acid supplementation, mainly with ω3 PUFAs, on exercise performance, because of their antioxidant effects [89]. As mentioned before, particular conditions of physical exercise, specifically those involving extremely intense exercise, very concentrated training, or periods of competition, as well as exercise sessions with a high predominance or proportion of eccentric contractions, can result in increased oxidative damage to cellular constituents. This is because such exercise increases the production of ROS/RNS in skeletal muscle above a physiological threshold [90]. Under these conditions, instead of being positive triggers of adaptive cellular signaling pathways, high levels of ROS cause functional oxidative damage to proteins, lipids, and other cell components that could exacerbate atrophy, sarcopenia, and myopathy factors in muscle. The persistence of greatly elevated levels of ROS at the local level may also reduce muscle reparation and differentiation of myoblasts and myotubes [91]. However, it has been shown that the MUFAs in chlorella reduce lipid peroxidation and oxidative stress damage [88]. Additionally, ω3 ALA is an antioxidant fatty acid that enhances eNOS activity and inhibits superoxide and peroxynitrite formation, thereby contributing to endothelial protection among other possible activities [92]. The effect of ω3 ALA on endothelial function can also be related to its activity on SIRT3 impairment, which in turn restores the mitochondrial redox balance in endothelial cells [93].

The antioxidant potential and free radical scavenging activity of ω3 ALA can also protect against cellular damage, apoptosis, and inflammation [94,95]. Inflammatory responses have been observed in athletes who engage in long-duration exercise, such as marathons or triathlons [96]. Inflammation is a physiological response to tissue damage that increases the expression of TNF-α, IL-1β, and IL-6 via NFķB. After muscle injury, NFķB increases the expression of RING-finger protein-1 (MuRF1), eventually promoting muscle wasting [97,98]. ω3 ALA intake seems to reduce plasma concentrations of IL-6 and other molecules related to inflammatory signaling, such as C-reactive protein, E-selectin, ICAM-1, and soluble vascular cell adhesion molecule 1 (VCAM-1), thereby helping to control the inflammatory response [99]. Several epidemiological studies show that ω6 LA also reduces the inflammatory response, despite its pro-inflammatory effect in vitro [100]. An adequate proportion of these two principal PUFAs (ω6 LA and ω3 ALA) in chlorella seems important to arrive at the desired anti-inflammatory effect, as a high ω6 diet can inhibit the anti-inflammatory and inflammation-resolving effects of the ω3 fatty acids [100].

A reduction of plasma IL-6 by ω3 ALA has also been reported during resistance training in older men [101]. This action may be based on the stimulation of muscle protein synthesis and enhancement of muscle mass via mTOR signaling, as described for ω3 PUFAs [102]. Insulin signaling plays a key role in mTOR activation, so PUFA supplementation might alleviate anabolic resistance [103] and improve the action of insulin, as skeletal muscle is the major site of insulin-stimulated glucose disposal [104]. Interestingly, metabolic benefits in terms of muscle glycemia have been reported for ω6 LA rather than ω3 PUFA [105], and there is an inverse association between the serum proportion ω6 LA and both fasting plasma glucose and post-load glucose [106]. ω3 ALA is also active at the metabolic level, as it increases the activity of carnitine palmitoyl transferase I and FA translocase [107]. The activity of these enzymes increases maximal fat oxidation in mitochondria, thus supporting the assumption that ω3 ALA intake may be used as a nutritional support to increase aerobic performance with substantial reliance on lipid-based metabolism and, at the same time, a glycogen sparing effect [107]. Dietary ω3 ALA and ω6 LA are oxidized much more rapidly than other biologically active fatty acids, such as docosahexaenoic acid (DHA) and arachidonic acid [108]. Specifically, ω3 ALA has been shown to have the highest rate of β-oxidation among the unsaturated fatty acids tested [86], so ω3 ALA could serve better than other fatty acids, as an energy substrate during long-term exercise bouts when carbohydrate reserves are depleted.

The dietary lipid profile determines the tissue phospholipid composition, which modulates not only insulin signaling but also the fluidity and rigidity of the cell membrane [109,110]. The administration of ω3 ALA maintains the integrity of the membrane in red blood cells (RBCs) when exposed to oxidative damage, and reduces lipid peroxidation [95]. The inclusion of ω3 PUFAs in the phospholipids of the RBC membrane increases the loss of deformability induced by exercise, thus improving performance by enhancing oxygen transport to the skeletal muscle [111]. However, most studies of ω3 PUFAs and RBC deformability have used fish oil, which is rich in ALA-derived ω3 PUFAs, mainly eicosapentaenoic acid (EPA) and docosahexaenoic acid (DHA), but not with the major fatty acids found in *Chlorella* (LA and ALA). Additionally, there is some concern about the adequate dosage of ω3 fatty acids, as excessive incorporation into plasma and tissue lipids may increase their susceptibility to lipid peroxidation, which is much more evident in athletes who may undergo high levels of oxidative stress [111]. So, further studies are still needed to provide better knowledge of the effects of fatty acids from *Chlorella* on cell membrane dynamics and the implications in the context of physical exercise.

## 5. Discussion

Chlorella is a low-fat, rich source of high-quality protein that has been authorized for human consumption by different regulatory agencies, including the European Food Safety Authority (EFSA). Chlorella also contains other bioactive macronutrients, such as fiber and unsaturated fatty acids. The proportion of each macronutrient stored in chlorella biomass can be maximized by modulating the growing conditions of this microalgae, which makes it an interesting option for the food industry as well as offering an opportunity to improve public health through better nutrition [6]. The intake of chlorella as part of a regular diet or as a supplement for humans and other mammals has been associated with many activities related to the improvement of exercise performance and recovery, such as glycemia and lipid balance and immunomodulation (Figure 1) [112].

In general, chlorella intake improves anaerobic and aerobic exercise performance as well as physical stamina and reduces the onset of fatigue [30,32,33,34]. One of the primary bases for all these benefits could be the antioxidant capacity of its macronutrients, such as peptides, polysaccharides, and fatty acids. This antioxidant capacity of chlorella also confers it an interesting anti-inflammatory capacity that could be useful in some exercise conditions and regimens [41]. Moreover, the effects of chlorella on endurance seem to be related to its activities at the metabolic level, improving glycemic control via the enhancement of muscle glucose uptake and sparing of glycogen stores, and augmenting blood flow to the muscles by increasing NO production and vasodilatation [34,35,45]. Chlorella is used in humans at doses of a few grams per day and is considered safe at doses of up to 10–15 g per day [16]. Cardiovascular benefits have been shown at a dose of 4 g per day, while a lower dose of 1.5 g per day was ineffective in patients with type-2 diabetes mellitus [113].

Besides the activities (antioxidant, anti-inflammatory, and metabolic) common to all the macronutrients in chlorella, each of them makes different specific contributions that improve exercise performance. The high content of protein in chlorella makes it a good dietary recommendation in physical exercise contexts, as it increases satiety, activation of the anabolic mTOR pathway in skeletal muscle, and the thermic effects of meals [53]. Proteins from *Chlorella* also increase the amount of intramuscular free amino acids and enhance the ability of the muscles to utilize them during moderate exercise, thereby sparing glycogen and improving the biochemical capacity of the muscle for fatty acid oxidation through β-oxidation [60,61]. The carbohydrates from *Chlorella* contribute to the benefits of dietary fiber, which promotes the right balance of gut microbiota and increases its diversity; two factors also related to the exercise performance of the host [73]. This prebiotic effect of chlorella also increases the production of SCFAs, which can be used as carbon and energy sources by liver and muscle cells, thus improving endurance by maintaining blood glucose levels [81]. Finally, lipids from *Chlorella*, mainly ω3 ALA and ω6 LA, can modify tissue phospholipid composition, contributing to the fluidity and rigidity of the cell membranes, which are important aspects of physical performance [111].

This review summarizes the still limited information available on chlorella and physical exercise in humans and other animals. The main strength of this work lies in the conclusion that all the studies reviewed underscore the positive effects of chlorella on physical performance. The main limitation of the review lies in the difficulty of establishing possible mechanisms of action, as chlorella is a complex mixture of bioactive components with different activities. More accurate mechanistic suggestions can be made when looking at individual components. Therefore, an important section of this review examines the role of proteins and peptides, polysaccharides (fiber), and unsaturated fatty acids, the main bioactive components of chlorella, whether or not they have been isolated from this and other microalgae. This is a limitation because it is always difficult to extrapolate such results to the whole mixture. Furthermore, the doses used in the different studies may differ considerably from those in supplements. Moreover, cooperative or opposite effects may be as important as those attributed to the individual components.

Despite the increasing evidence linking chlorella consumption to the improvement of different aspects of physical exercise, there is a lack of consensus about the significance of the different mechanism(s) proposed. The observations made related to the bioactivity of chlorella in the context of physical exercise require verification by new mechanistic studies in animal models. In addition, research is needed on adequate dosages to be administered under different exercise conditions and regimens and to ascertain if a personalized prescription is required. Moreover, as the efficacy of chlorella supplementation on gut microbiota will also depend on the previous status of the host, which is in turn modulated by his or her engagement in physical exercise, new studies should address these complex interrelationships. As microalgae, particularly chlorella, are rich sources of high-quality proteins and peptides, they are good candidates for functional components in sports nutrition, while it is also being demonstrated that other components in chlorella contribute to sports performance. In particular, the cell wall is a source of prebiotic polysaccharides. Prebiotics help to maintain a balanced intestinal microbiota and are important actors in the prevention of non-communicable diseases such as diabetes. Recently, beneficial effects of exercise on the gut microbiota have been revealed [114], and still more recent studies are exploring the inverse relationship: the influence of the gut microbiota on exercise performance via the action of gut fermentation-derived short-chain fatty acids and the stimulation of various pathways of lactate metabolism (same reference). Polysaccharides in chlorella preparations are released from the cell wall by enzymatic treatments. To maximize the prebiotic potential of chlorella-derived products, future efforts should be devoted to the optimization of hydrolysis conditions.

The evidence available to date shows that chlorella, as well as spirulina and maybe other microalga, can effectively be used as supplements or even meal replacements to improve sports performance or simply to maximize the benefits of moderate exercise in the normal population. Microalgae may be one of the foodstuffs of the future, since they are a source of high-quality protein and many other bioactive components, and they may also significantly contribute to a sustainable world through their contribution to the fixation of carbon dioxide and a reduction in the amount of land used for the production of animal feed.

## 6. Conclusions

Chlorella improves anaerobic and aerobic exercise performance, physical stamina, and reduces the onset of fatigue. These benefits could be attributed to the antioxidant, anti-inflammatory, and metabolic activities common to its macronutrients (peptides, polysaccharides, and fatty acids). Moreover, each macronutrient brings to Chlorella some particular functions that could be related to the improvement of physical exercise and performance.

Proteins from *Chlorella* increase the amount of intramuscular free amino acids and enhance the ability of the muscles to utilize them, thereby sparing glycogen and improving the biochemical capacity for fatty acid β-oxidation. Some carbohydrates derived from *Chlorella* cell walls, help to maintain a balanced gut microbiota and increase its diversity. This prebiotic effect of chlorella results in an increased production of SCFAs, which can be used as carbon and energy sources. Thus, carbohydrate-derived SCFAs improve endurance by maintaining blood glucose levels. Finally, lipids such as ω3 ALA and ω6 LA can contribute to physical performance by favorably modifying the fluidity and rigidity of the cell membranes.

## Figures and Tables

**Figure 1 nutrients-15-02168-f001:**
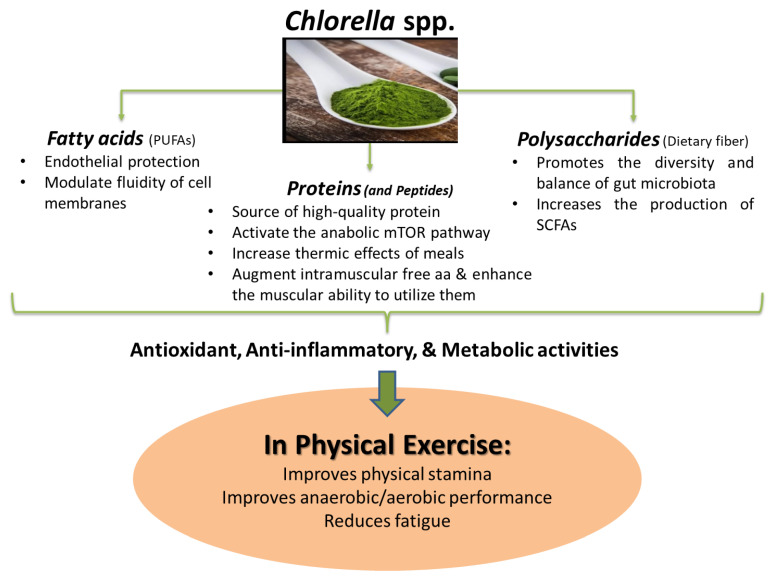
Effects of *Chlorella* and its macronutrients on physical exercise.

**Table 1 nutrients-15-02168-t001:** Most relevant biochemical composition of *Chlorella*.

	Main Components
**Macronutrients**	
Proteins	All the essential amino acids
Carbohydrates	α- and β-glucans, fiber
Lipids	PUFAs (linoleic acid, alpha-linolenic acid)
**Micronutrients**	
Minerals	Na^+^, K^+^, Fe^2+^, Ca^2+^, Mg^2+^, Mn^2+^, Zn^2+^, Co^2+^
Vitamins	A, B_1_, B_2_, B_6_, B_12_, C, E, K_1_, folic and pantothenic acids, niacin
Pigments	Chlorophylls, carotenoids, lutein

**Table 2 nutrients-15-02168-t002:** Studies involving the effects of *Chlorella* administration on physical parameters.

Subjects	Supplementation Protocol	Exercise Protocols and Tests	Results
ICR mice, male Healthy	*C. vulgaris* (Hot water extract) Dosage: 0.05–0.15 g/kg/day Intervention: 1 week	Training: -- Forced swimming test (6′, measured the total duration of immobility.	↓ Immobility time, BUN, CK, and LDH No effect: Glc, TP [30]
ICR mice, male 4 weeks old Healthy	*C. vulgaris* (Hydrolyzed) Dosage: 10 mL/kg/day Intervention: 2 weeks	Training: -- Forced swimming test (6′, measured the total duration of immobility.	↑ IFN-γ, IL-2 (in Molt-4 cells) ↓ Immobility time, BUN [31]
BALB/c mice, male 6 weeks old Healthy	Chlorella powder in chow Dosage: 0.5%, 1 mg/kg/day Intervention: 2 weeks	Training: -- Forced swimming test (Measured the maximum swimming time).	↑ Swimming time, FFA, Glc, TG, and lactic acid. ↓ Oxidoreductase activity and the leukotriene synthesis pathway [32]
Men and women ≈21 years old Healthy	Chlorella tablets Dosage: 15 tablets, 2 times/day Intervention: 4 weeks	Training: -- Maximal exercise test (Incremental cycling to exhaustion).	↑ VO_2_ peak [33]
Sprague-Dawley rats, male 12 weeks old Healthy	Chlorella powder in chow Dosage: 0.5% Intervention: 6 weeks	Training: HIIE (14 × 20″ swim, with a 10″ pause between series, bearing a weight, 4 days/week, for 6 week). Exercise performance test (Maximal number of HIIE).	↑ number HIIE sessions, the expression of MCT1, MCT4, and PPARγ coactivator-1α, and the activities of LDH, CS, and COX in the red region of gastrocnemius. No effect: MCT1 expression and LDH, CS, and COX activities in the white region of gastrocnemius [34]
OLETF rats, male 20 weeks old Type 2 diabetics	Chlorella powder in chow Dosage: 0.5% Intervention: 8 week	Training: Aerobic exercise (running on the treadmill for 1 h, 25 m/min, 5 days/week, during 8 weeks).	↑ insulin sensitivity index concomitant with muscle PI3K activity, Akt phosphorylation, and GLUT4 translocation levels ↓ fasting blood glucose, insulin levels, and total glucose AUC during the OGTT [35]
Men ≈23 years old Overweight	*C. vulgaris* tablets Dosage: 300 mg, 4 times/day Intervention: 1 week	Training: -- Acute eccentric exercise protocol (20´ treadmill run at a speed of 9 km/h with a negative 10% slope, 1 week after supplementation)	↓ IL-6 levels and insulin resistance, 24 h after acute eccentric exercise test [36]

Legend. HIIE: high-intensity intermittent exercise; MCT: monocarboxylate transporter; LDH: lactate dehydrogenase; CS: citrate synthase; COX: cytochrome-c oxidase; ICR: Institute of Cancer Research; CVE: extracts of *C. vulgaris*; BUN: blood urea nitrogen; CK: creatine kinase; Glc: glucose; TP: total protein; HCV: *Chlorella vulgaris* by malted barley; CPK: creatine phosphokinase; IFN-γ: interferon gamma; IL-2: interleukin 2; OLETF: Otsuka Long-Evans Tokushima Fatty; AUC: area under the curve; OGTT: oral glucose tolerance test; PI3K: phosphatidylinositol-3 kinase; Akt: protein kinase B; GLUT4: glucose transporter 4; BALB/c: albino, laboratory-bred strain; FFA: free fatty acid; TG: triglyceride; HR: heart rate; PPAR: peroxisome proliferator-activated receptor.

## Data Availability

Not applicable.

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
