# Peer review of "Bioactivity of Macronutrients from Chlorella in Physical Exercise"

_nutrients, 2023, doi:10.3390/nu15092168_

Round 1

Reviewer 1 Report

I would like to congratulate the authors for the structure of the manuscript and all the research carried out. However, there are some concerns, in part important, so the review articles need revision, see below.

Introduction

·       Why is this study considered relevant?

·       I suggest you incorporate a little more information related to Chlorella in relation to other clinical uses in humans.

·       Which herbal supplements have similar properties to Chlorella?

·       describe potential side effects

Methods

·       Include a small part of the methodology: databases used, search terms, inclusion/exclusion criteria.

Results

·       It should include a table and figure of the findings described.

Discussion

·       Include a section on strengths / limitations.

·       Is it possible to describe more mechanisms responsible for the described actions?

·       What does this article contribute to, the authors should make their own assessment and include their own discussion of the results shown in the manuscript?

·       Does it apply to all types of exercise? or only to aerobic exercise?

·       Does it prevent the absorption of other nutrients needed for exercise?

·       Is there an effective? and toxic dose range? when is supplementation recommended?

·       include a section on future scenarios

Conclusion

·        In the Conclusion section, state the most important outcome of your work. Do not simply summarize the points already made in the body — instead, interpret your findings at a higher level of abstraction. Show whether, or to what extent, you have succeeded in addressing the need stated in the Introduction (or objectives).

Author Response

Reviewer 1

I would like to congratulate the authors for the structure of the manuscript and all the research carried out. However, there are some concerns, in part important, so the review articles need revision, see below.

Introduction

  • Why is this study considered relevant?

This aspect has been included this consideration in lines 142-148.

  • I suggest you incorporate a little more information related to Chlorella in relation to other clinical uses in humans.

We have added this information in lines 90-95.

  • Which herbal supplements have similar properties to Chlorella?

We do not comment on any other supplement as preparations of chlorella and spirulina are quite different from herbal supplements, which are usually extracts with low protein content. Nevertheless, we have expanded the information about spirulina in Lines 75-77.

  • describe potential side effects

Side effects of Chlorella intake are exposed in lines 86-90.

Methods

  • Include a small part of the methodology: databases used, search terms, inclusion/exclusion criteria.

Despite this revision is not a systematic one, a brief ‘Methods’ section has been included following the reviewer suggestion.

Results

  • It should include a table and figure of the findings described.

The manuscript now includes the new Table 2 and the Figure 1.

Discussion

  • Include a section on strengths / limitations.

The strengths / limitations of this review are now described in lines 484-495.

  • Is it possible to describe more mechanisms responsible for the described actions?

This is a point still open for future research and our group is hoping to make future contributions in this area.

  • What does this article contribute to, the authors should make their own assessment and include their own discussion of the results shown in the manuscript?

Now the last section of the manuscript includes the Discussion and Conclusion of the revision process.

  • Does it apply to all types of exercise? or only to aerobic exercise?

The text includes studies related to both aerobic and anaerobic exercise as already stated in the abstract and in the main text.

  • Does it prevent the absorption of other nutrients needed for exercise?

So far there is no available information concerning any possible influence of chlorella on the absorption of other nutrients.

  • Is there an effective? and toxic dose range? when is supplementation recommended?

These data are now included in lines 485-489.

  • include a section on future scenarios

The future scenarios are now described in lines 466-468.

 Conclusion

  • In the Conclusion section, state the most important outcome of your work. Do not simply summarize the points already made in the body — instead, interpret your findings at a higher level of abstraction. Show whether, or to what extent, you have succeeded in addressing the need stated in the Introduction (or objectives).

This change is now included in lines 504-515.

Reviewer 2 Report

This review has been very well written and has significant content contributing to the topic. In addition, I have raised some small suggestions in order to improve the manuscript.

L29 – All acronyms must be previously named in their first mention.

L137 – “spp” should not be in italics

General comments

The review lacks a session or paragraph indicating how the authors performed the search, the databases used, the inclusion and exclusion criteria, how many papers were retrieved, how many were chosen, and so on.

There is an odd contrast between large paragraphs versus tiny ones. The authors should double-check and split some large paragraphs to enhance conciseness and cohesion.

I miss more tables and figures in the review. Generally, review articles present schemes, charts, and other related figures to illustrate better and makes the reading more visually pleasant. For instance, the authors could present a table summarizing the main effects of Chlorella macronutrients on physical exercise, as this is the main objective of the review. Also, an image illustrating the main effects reported in the literature on the human body could also fit well.

Author Response

Reviewer 1

I would like to congratulate the authors for the structure of the manuscript and all the research carried out. It is highly publishable. The authors have substantially improved the manuscript with post-revision changes. However, there are some concerns, in part important, so the review articles need revision, see below.

Results

  • I suggest improving table 2 because it is difficult to interpret the most relevant findings, you can use abbreviations or symbols to highlight the results

Following the reviewer suggestion, e have improved the Table 2.

Round 2

Reviewer 1 Report

I would like to congratulate the authors for the structure of the manuscript and all the research carried out. It is highly publishable. The authors have substantially improved the manuscript with post-revision changes. However, there are some concerns, in part important, so the review articles need revision, see below.

Results

·       I suggest improving table 2 because it is difficult to interpret the most relevant findings, you can use abbreviations or symbols to highlight the results

Author Response

(The authors gave the same response as above.)
